# RETROSPECTIVE ATTENTION SMOOTHING: CORRECTING CAUSAL BIAS IN AUTOREGRESSIVE LANGUAGE MODELS

## ABSTRACT

Large Language Models (LLMs) often exhibit *contextual faithfulness hallucinations*, producing outputs that deviate from the intended meaning of the full input. One contributing factor is the causal masking mechanism, which restricts the model to prefix information and may lead to biased or incomplete semantic representations. To address this, we propose **Retrospective Attention Smoothing (RAS)**, a framework that retrospectively refines hidden representations. RAS models hidden states as an *absorbing Markov chain (AMC)* in semantic space, where the final hidden state represents the semantics of the complete input. By analyzing possible semantic trajectories, AMC provides a natural measure of *semantic surprise*, signaling where prefix interpretations diverge from the whole context. These signals guide a smoother that modifies only query vectors, bridging past and future semantics so that future information can be integrated into earlier representations. To adapt RAS to each input, we introduce a lightweight *retrospective adaptation* procedure balancing language modeling accuracy, query stability, and surprise minimization. Experiments on multiple QA benchmarks show that RAS consistently reduces hallucinations, offering an innovative way to enhance the semantic faithfulness of LLMs without altering the frozen backbone.

## 1 INTRODUCTION

Large Language Models (LLMs) have demonstrated remarkable capabilities across a wide range of natural language processing tasks (Li et al., 2024b; Zhang et al., 2023a; Ravaut et al., 2024; Min et al., 2023; Peng et al., 2023; Demszky et al., 2023). Beyond surface-level performance, recent research has shown that LLMs can internally summarize, process, and propagate information through structured mechanisms, such as inductive heads, anchor tokens, and interpretable circuits (Olsson et al., 2022; Wang et al., 2023; 2025). These findings suggest that LLMs do not merely memorize correlations, but maintain latent representations that capture semantic trajectories across the input sequence.

However, despite such abilities, LLMs remain prone to contextual hallucinations (Zhang et al., 2023b; Tonmoy et al., 2024; Pan et al., 2024)—producing outputs that deviate from the intended meaning of the full input. A central reason lies in the *causal masking mechanism*: during inference, the model is restricted to prefix information, and thus its hidden representations may reflect biased or incomplete semantics. For instance, as shown in Figure 1, when asked *"What is the human body's largest organ?"*, a prefix-limited model often defaults to *"the liver,"* guided by the common intuition that *"largest organ"* refers to an internal organ. This guess appears plausible—until the query continues: *"...and based on that organ, which vitamin is synthesized upon sunlight exposure, and name a deficiency disease caused by its lack?"* At this point the model encounters *surprise*: its earlier prediction clashes with the new evidence. Still biased, it may force a continuation such as *"Vitamin A; night blindness,"* even though the correct reasoning path is *skin → Vitamin D → rickets*.

This example illustrates how prefix-induced bias can cascade into multi-hop reasoning errors, yet also how intermediate cues (e.g., *sunlight exposure*) implicitly point toward the correct semantics. To formalize this, we model semantic evolution as an *absorbing Markov chain (AMC)*: each state represents a transient semantic representation along the pathway, and the final state—after the model

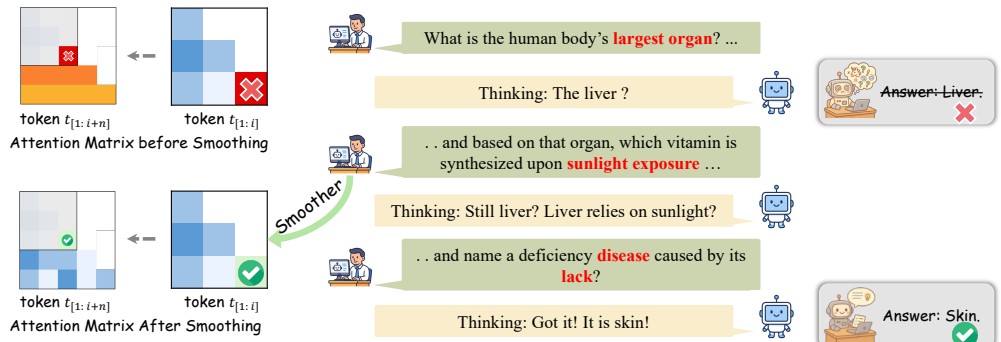

Figure 1: Illustration of **Retrospective Attention Smoothing (RAS)**. **Top:** Without smoothing, attention remains biased toward early prefixes, leading to the incorrect answer *liver* despite later cues (*sunlight exposure*, *disease*). **Bottom:** With RAS, queries are retrospectively adjusted so that future information is integrated into earlier semantics, enabling the model to re-align reasoning and reach the faithful answer *skin*.

processes the entire input—serves as the absorbing state. AMC does not directly reveal the "correct" answer, but it quantifies how easily prefix semantics can transition toward the semantics of the complete input. Crucially, this formulation allows us to retrospectively identify where early pathways diverge from the full semantics, and to use these signals to *directly reweight attention* and reshape the semantic pathway itself. This motivates our method: **Retrospective Attention Smoothing (RAS)**.

**How does RAS work?** RAS leverages absorbing Markov chains to model multiple possible *semantic pathways* in semantic space. By examining how easily early states can transition toward the semantics of the complete input, we obtain a natural measure of *semantic surprise*: unlikely or circuitous transitions indicate biased interpretations. These AMC-derived signals then guide a *parameter-free, two-pass* adjustment: in a second, non-causal pass on selected layers and heads, we reweight attention using the semantic signals (and their utilization in the first pass) and fuse the corrected attention output with the original masked output. This *zero-training* procedure establishes a bridge between past and future semantics so that future information can be effectively integrated into earlier interpretations, helping attention focus on the most relevant parts of the upcoming context and correcting prefix-induced bias.

**Paper roadmap.** We first introduce a quantitative framework for evaluating semantic consistency between prefixes and complete inputs via AMC (including pathway scores and semantic surprise). We then detail *Retrospective Attention Smoothing*: how AMC-derived signals are computed, how they interact with utilization to reweight attention in a second pass, and how the corrected attention is fused with the original decoding stream without updating model parameters. Finally, we evaluate our approach on multiple QA benchmarks, showing that RAS consistently reduces hallucinations and improves exact match and F1 scores.

**Our contributions are threefold:**

- We quantify prefix-induced semantic bias in semantic space using absorbing Markov chains, yielding pathway-based measures of semantic surprise.

- We propose *Retrospective Attention Smoothing (RAS)*, a *training-free*, two-pass attention reweighting mechanism guided by AMC-derived semantic signals that integrates future semantics into earlier interpretations.

- We empirically validate RAS across multiple QA benchmarks, demonstrating substantial improvements in semantic faithfulness and answer accuracy without modifying the frozen backbone.

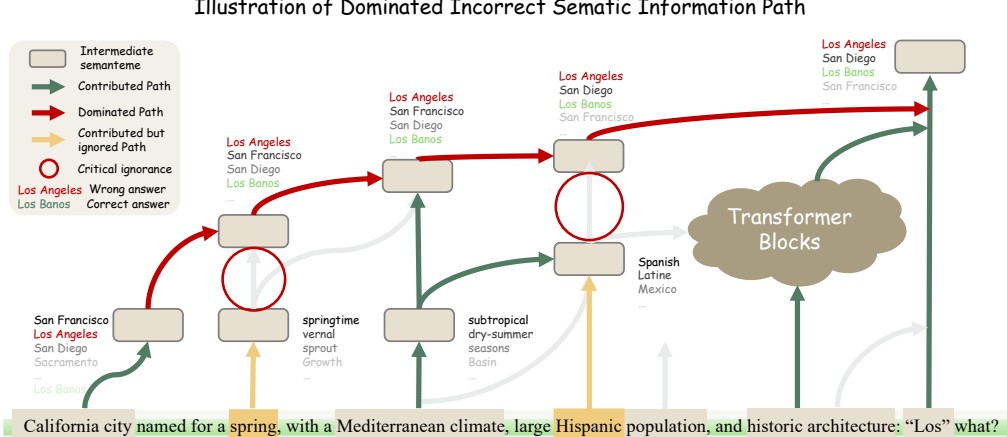

Figure 2: Illustration of a dominated incorrect semantic path in LLM decoding. Although the correct answer is *Los Banos* (green path), the model prediction follows a dominated path (red) leading to *Los Angeles*. Yellow arrows indicate ignored correct paths, and red circles mark critical points of semantic neglect.

## 2 RELATED WORK

### 2.1 HALLUCINATIONS IN LLMS

Recent studies have shown that hallucinations in LLMs can take many forms, and in particular we focus on what we call *contextual faithfulness hallucinations*—cases where the generated content diverges from the meaning of the full input. Such hallucinations may arise for several reasons, including exposure to massive training data that contains fabricated, outdated, or biased information (Zhang et al., 2023b). The versatility of LLMs across tasks, languages, and domains further complicates both their evaluation and mitigation (Tonmoy et al., 2024). A wide range of approaches have been explored, such as Retrieval-Augmented Generation (RAG) (Lewis et al., 2020), inference-time interventions (Li et al., 2024a), external knowledge retrieval (Varshney et al., 2023), self-reflection (Ji et al., 2023), uncertainty estimation (Lin et al., 2023), chain-of-thought prompting (Wei et al., 2022), and system-level prompting (Touvron et al., 2023). These techniques share the goal of grounding model outputs in factual information and maintaining stronger alignment with the semantic pathways present in the input. However, most existing methods operate primarily at the *output level*, aiming to steer final predictions toward truthfulness. In contrast, our approach directly targets the *semantic pathways* inside the model, retrospectively adjusting them to reduce prefix-induced bias and improve contextual faithfulness. Despite these advances, hallucination remains a persistent challenge, motivating further innovation and evaluation (Zhang et al., 2023b; Tonmoy et al., 2024).

### 2.2 CONSTRAINED DECODING STRATEGIES

Another line of research addresses hallucinations through decoding-time interventions, as modifying model parameters directly is computationally costly. For example, Context-Aware Decoding (CAD) uses a contrastive distribution to amplify differences between outputs with and without the guiding semantic pathways, thereby overriding misleading priors (Shi et al., 2023). Inference-Time Intervention (ITI) shifts model activations during inference by targeting attention heads with high probing accuracy for truthfulness (Li et al., 2024a). Decoding by Contrasting Layers (DOLA) compares logits from earlier and later layers to suppress incorrect facts (Chuang et al., 2023). Activation Decoding manipulates activation patterns by optimizing the sharpness of in-context activations, guiding the model toward more faithful semantic pathways (Chen et al., 2024). Collectively, these decoding strategies steer generation toward more reliable and contextually faithful results without retraining the backbone model. However, they remain focused on constraining the output distribution during decoding, rather than directly modeling and adjusting the semantic pathways in semantic space as we propose.

## 2.3 SEMANTIC PATHWAYS IN LLMs

A growing body of research has investigated how semantic information propagates inside LLMs, seeking to uncover the pathways through which evidence is aggregated and transformed. Abnar & Zuidema (2020) proposed attention rollout and attention flow to better approximate token relevance, showing that raw attention weights alone can be misleading. Ferrando & Voita (2024) traced semantic pathways by identifying influential nodes and edges in a forward pass. Wang et al. (2023) showed that label words in in-context learning act as anchors, aggregating information from demonstrations in shallow layers before guiding predictions in deeper layers. Yao et al. (2024) uncovered knowledge circuits by identifying key attention heads and MLPs that jointly encode factual knowledge. Yuan et al. (2021) proposed Transition Attention Maps, combining Markov chains with integrated gradients to track token relevance across layers.

Together, these works highlight the importance of modeling semantic pathways for understanding and improving the reliability of LLM reasoning. Our approach builds on this perspective but goes a step further: rather than focusing only on token-level attribution, we model semantic pathways themselves as an absorbing Markov chain, enabling us to capture how intermediate semantics connect to the complete input and to use this structure for retrospective attention smoothing.

## 3 PRELIMINARY

Before introducing our method, we briefly review the foundations of autoregressive language models and absorbing Markov chains. This provides the necessary background for understanding how we later formulate semantic pathways in semantic space and use them to design Retrospective Attention Smoothing (RAS).

### 3.1 LANGUAGE MODEL ARCHITECTURE

An autoregressive language model generates text by predicting the next token conditioned on the sequence of previously observed tokens. Formally, given a sequence $x_1, x_2, \ldots, x_t$, the next-token distribution is modeled as:

$$\mathbb{P}(x_{t+1} \mid x_1, x_2, \ldots, x_t). \tag{1}$$

Modern LLMs typically implement this distribution using the Transformer architecture, where representations are updated layer by layer and contextualized through attention. While this formulation ensures that every prefix in principle contributes to the prediction of subsequent tokens, in practice, training biases and the causal mask can cause the model to rely disproportionately on local or frequent cues. This leads to what we refer to as *contextual faithfulness hallucinations*: the model's prediction is consistent with the prefix but diverges from the semantics of the complete input.

To better characterize and mitigate this issue, we shift the perspective from token-level probabilities to *semantic pathways*: trajectories in semantic space that summarize the evolving meaning of prefixes. By considering not only the immediate prefix but also the potential pathways connecting intermediate semantics to the semantics of the complete input, we can formally capture where biases occur and how to correct them. This motivates our use of absorbing Markov chains as a mathematical framework for modeling semantic pathways.

### 3.2 ABSORBING MARKOV CHAIN FORMULATION

We now recall the basics of absorbing Markov chains (AMC) and explain how they provide a natural tool to analyze semantic pathways in LLMs.

Let $\Omega$ denote a finite state space with $|\Omega|$ elements. A discrete time-homogeneous Markov chain is defined as $X(\Omega, Q)$ with state space $\Omega = \{x_i\}_{i=1}^{|\Omega|}$ and transition matrix $Q \in \mathbb{R}^{|\Omega| \times |\Omega|}$, where $Q_{ij} = Q(x_i, x_j)$ represents the probability of moving from state $x_i$ to state $x_j$. A Markov chain is a sequence of random variables $X = (X_1, X_2, \ldots)$ that satisfies the Markov property:

$$P(X_{n+1} = x_{n+1} \mid X_1 = x_1, \ldots, X_n = x_n) = P(X_{n+1} = x_{n+1} \mid X_n = x_n)$$
$$:= Q(x_n, x_{n+1}).$$

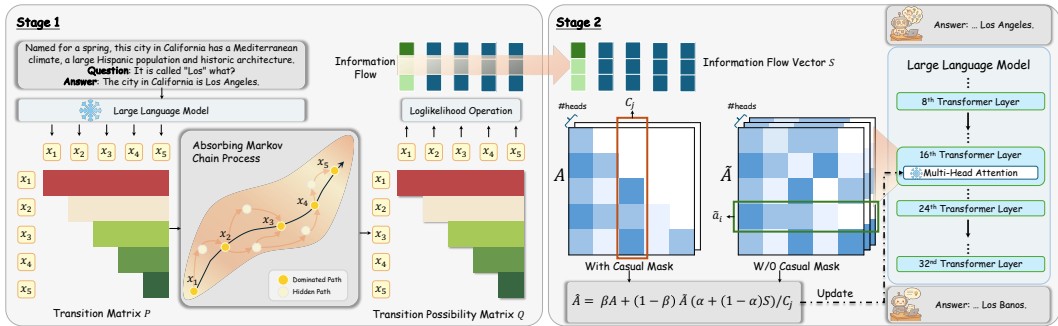

Figure 3: **Overall pipeline of our absorbing Markov chain (AMC) model. Stage 1:** a frozen LLM produces token-level likelihoods, which are converted into a causal transition matrix and normalized into an AMC matrix $Q$; from $Q$ we extract semantic pathways and compute a *Information Score* vector $\mathbf{s}$ that measures how later tokens should retrospectively influence earlier ones. **Stage 2:** at a selected Transformer layer, the original masked attention $A$ is interpolated with an AMC-guided unmasked attention $\tilde{A}$ reweighted by $\mathbf{s}$, producing a final attention map that integrates future semantics into earlier queries. This training-free, two-pass adjustment reduces prefix bias and mitigates contextual faithfulness hallucinations.

An absorbing Markov chain is a Markov chain that contains at least one *absorbing state*, i.e., a state that, once entered, cannot be left. States that are not absorbing are called *transient*. If an absorbing Markov chain has $r$ absorbing states and $t$ transient states, its transition matrix $\tilde{P}$ can be written in canonical form as:

$$\tilde{P} = \begin{bmatrix} Q & R \\ \mathbf{0} & I_r \end{bmatrix}, \tag{2}$$

where $Q$ is a $t \times t$ matrix representing transitions among transient states, $R$ is a $t \times r$ matrix for transitions from transient to absorbing states, $\mathbf{0}$ is an $r \times t$ zero matrix, and $I_r$ is an $r \times r$ identity matrix.

In our formulation, the transient states correspond to intermediate semantics along the prefix pathway, and the absorbing state corresponds to the semantics after observing the complete input. Modeling semantic pathways as an AMC allows us to analyze the probability of different trajectories and to quantify *semantic surprise* when a prefix pathway poorly aligns with the eventual semantics revealed by the full input. This insight forms the basis for our Retrospective Attention Smoothing (RAS) method.

## 4 METHODOLOGY

### 4.1 SEMANTIC PATHWAYS AS ABSORBING MARKOV CHAINS

We view the evolution of semantics in a sequence as a pathway in an absorbing Markov chain (AMC). Given an input with $T$ tokens $z = (z_1, z_2, \ldots, z_T)$, we define an absorbing chain $X_z(\Omega_z, \tilde{P}_z)$ starting at $z_1$ and ending at $z_T$, where the state space is

$$\Omega_z := \{z_i\}_{i=1}^{T}.$$

The canonical transition matrix is

$$\tilde{P}_z = \begin{bmatrix} Q_z & R_z \\ \mathbf{0} & 1 \end{bmatrix}, \tag{3}$$

where $Q_z \in \mathbb{R}^{(T-1) \times (T-1)}$ encodes transitions among transient semantic states, $R_z \in \mathbb{R}^{(T-1) \times 1}$ encodes transitions into the absorbing state, $\mathbf{0}$ is a zero row vector, and $1$ represents the absorbing identity. Due to causal masking, $Q_z$ is upper triangular, ensuring that semantic pathways always progress forward.

**Quantifying semantic pathways.** To evaluate whether a pathway is faithful to the full input, we adapt the notion of cover time. For a Markov chain, the cover time $\tau$ is the first step when all states are visited:

$$\tau = \inf\{k \in \mathbb{N} \mid \Omega_z \subseteq (X_1, \ldots, X_k)\}.$$

In an AMC, $\mathbb{E}[\tau] = \infty$, since once the absorbing state is reached, the process halts. We therefore define *covering rate*:

$$r(z) := \mathbb{E}\left[\frac{T}{\tau}\right] \tag{4}$$

$$= \prod_{i=1}^{T-1} \tilde{P}_z(z_i, z_{i+1}). $$

This value is finite only if the process follows the exact order $(z_1, \ldots, z_T)$. Taking logarithms gives

$$\log r(z) = \sum_{i=1}^{T-1} \log \tilde{P}_z(z_i, z_{i+1}), \tag{5}$$

where low values indicate the divergence between prefix and full-input semantics, corresponding to high *semantic surprise*.

**Fundamental matrix.** The AMC is also characterized by its *fundamental matrix*:

$$N = (I - Q_z)^{-1}. \tag{6}$$

Here $N_{ij}$ is the expected number of visits to state $j$ starting from state $i$. Thus $N$ compactly encodes how early semantics connect to later ones and serves as the basis for computing our information score.

## 4.2 DYNAMIC ATTENTION ADJUSTMENT BASED ON SEMANTIC PATHWAYS

The core idea is to use AMC-derived signals to guide a second-pass correction of attention. From Stage 1, we compute the *Information Score* vector or information score $\mathbf{s} \in \mathbb{R}^T$:

$$s_j = -\log H_{1j}, \quad j = 1, \ldots, T, \tag{7}$$

where

$$H := (N - I_t)(N_{\text{dg}})^{-1}, \tag{8}$$

$\text{diag}(N)$ represents the diagonal matrix of $N$, and $H_{1j}$ denotes the normalized absorption flow from the initial state to semantic state $j$.

Intuitively, a large $s_j$ means token $j$ introduces substantial new semantics; such tokens are influential but also risky if their semantics are biased.

Performing a direct reduction of the coverage rate $r(z)$ during inference is intractable. Instead, we design a heuristic adjustment: tokens with high surprisal $s_j$ or high utilization should have moderated impact on attention. Formally, let $A \in \mathbb{R}^{H \times T \times T}$ denote the masked attention matrix (first pass), and $\tilde{A}$ the unmasked attention matrix (without causal mask). We compute the utilization vector $\mathbf{c} \in \mathbb{R}^T$ as

$$c_j = \sum_{i=1}^{T} A_{ij}, \quad j = 1, \ldots, T, \tag{9}$$

measuring how much attention token $j$ already received.

We then re-weight $\tilde{A}$ using surprisal $\mathbf{s}$ and utilization $\mathbf{c}$:

$$\tilde{A}' = \tilde{A} \odot \frac{\alpha + (1 - \alpha)\mathbf{s}}{\mathbf{c}}, \tag{10}$$

where $\odot$ and the division are element-wise and $\alpha \in [0, 1]$ interpolates between the original weighting $\tilde{A}$ and the AMC-guided weighting. Finally, we fuse outputs from masked and adjusted attention:

$$\hat{A}V = \beta(AV) + (1 - \beta)(\tilde{A}'V), \tag{11}$$

with $V$ is the matrix of attention values and $\beta \in [0, 1]$ balancing stability vs. correction.

This two-pass procedure retrospectively integrates future semantics into earlier queries, redistributes attention away from risky tokens, and heuristically reduces the effective coverage rate, thus mitigating contextual faithfulness hallucinations.

## 5 EXPERIMENTS

### 5.1 EXPERIMENTS SETUP

**Datasets.** We validate the effectiveness of the model on two tasks: hallucination detection and multi-choice question answering. For hallucination detection, we use the HaluEval QA dataset Li et al. (2023), which contains 10K hallucinated samples annotated by human labelers to evaluate the model's ability to recognize and avoid generating hallucinations. For multi-choice question answering, we use the WIKI-FACTOR and NEWS-FACTOR datasets from the FACTOR benchmark Muhlgay et al. (2023). WIKI-FACTOR is based on the Wikipedia section of The Pile's validation split and consists of 2994 examples, while NEWS-FACTOR is based on Reuters articles and consists of 1036 examples. These datasets are designed to test the model's factual reasoning and comprehension abilities.

**Evaluation Metrics.** For HaluEval, WIKI-FACTOR, and NEWS-FACTOR, we use precision as the evaluation metric for hallucination discrimination and performance of factual reasoning. The TruthfulQA-based open-ended text generation evaluation in Lin et al. (2021) is excluded, as the GPT-based judging setup used in prior work is no longer available.

**Models.** We use LLAMA2-7B-chat and LLAMA2-13B-chat Touvron et al. (2023) as the base models for evaluation. We also introduce the LLaMA3-8B-Instruct-Instruct Dubey et al. (2024) for comparison, in order to assess the effectiveness of the proposed approach on newer large language models.

**Baselines.** We compare our model to the following baselines: (1) original Decoding (or greedy decoding); (2) Dola Chuang et al. (2023), which subtracts the final layer logits from the logits of the earlier contrast layer to get the adjusted probability distribution of the next word; (3) Activation Decoding (AD) Chen et al. (2024), that adjusts the probability distribution of the next word according to the sharpness degree of the activation of the next token candidate. Each of these baselines uses only internal representations of the model to help decode and mitigate hallucinations, without the need for external information and extra training.

### 5.2 INFO SCORE DISTRIBUTION ANALYSIS

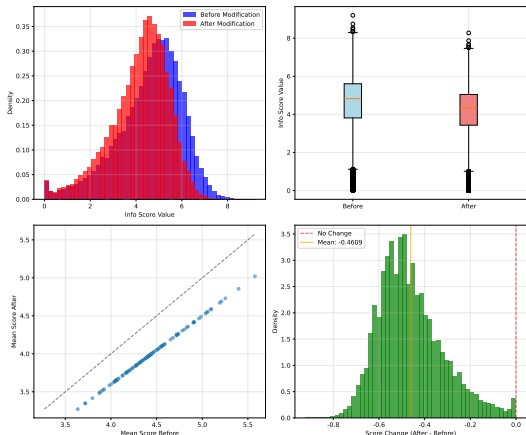

Figure 4: Info score distribution across HotpotQA samples. Top: token-level distributions (left) and box plot comparison (right). Bottom: sample-level mean score scatter (left) and change histogram (right).

We analyze the impact of our modification on the distribution using HotpotQA Yang et al. (2018) samples. As shown in Figure 4, token-level histograms reveal a leftward shift of scores from 4–6 to 3–5 after modification, while box plots confirm a median drop (5.0 → 4.0) and reduced variance, indicating more controlled allocation of attention. At the sample level, the scatter plot shows all points below the diagonal, evidencing consistent reductions in mean scores, with their linear arrangement suggesting proportional preservation across samples. Finally, the change histogram illustrates that most differences fall between $-0.4$ and $-0.6$, with a mean of $-0.46$, forming a stable and approximately Gaussian distribution. Overall, these results demonstrate that our approach systematically suppresses over-confident attention signals while maintaining proportional consistency across contexts, thereby mitigating context fidelity hallucinations and enhancing robustness in multi-hop reasoning tasks.

## 5.3 Main Results Analysis

| Model | TruthfulQA (MC) | | | FACTOR | | HaluEval |
|---|---|---|---|---|---|---|
| | MC1 | MC2 | MC3 | News | Wiki | QA |
| LLaMa2-7B-chat | 33.6 | 51.3 | 24.9 | | | |
| + Dola (Chuang et al., 2023) | 29.7 | 51.8 | 21.6 | 48.1 | **56.5** | 51.3 |
| + AD (Chen et al., 2024) | 34.0 | 51.6 | 25.8 | 61.7 | 53.8 | 52.4 |
| + Ours | **34.5** | **54.3** | **26.7** | **64.9** | 56.5 | **53.3** |
| LLaMa2-13B-chat | 35.0 | 53.3 | 26.6 | | | |
| + Dola (Chuang et al., 2023) | 27.1 | 45.8 | 22.9 | 50.6 | 49.1 | 49.4 |
| + AD (Chen et al., 2024) | 34.0 | 53.5 | 26.6 | 67.8 | 58.4 | 49.0 |
| + Ours | **35.8** | **56.5** | **28.1** | **69.3** | **60.9** | **50.1** |
| LLaMa3-8B-Instruct | 40.8 | 59.4 | 31.7 | | | |
| + Dola (Chuang et al., 2023) | 34.4 | 53.8 | 24.9 | 60.3 | **55.7** | 35.9 |
| + AD (Chen et al., 2024) | 33.9 | 56.9 | 28.9 | 59.9 | 48.2 | 35.7 |
| + Ours | **41.8** | **60.3** | **33.9** | **65.2** | 52.8 | **36.4** |

Table 1: Performance comparison on TruthfulQA (MC), FACTOR, and HaluEval datasets.

Table 1 demonstrates that our method consistently outperforms strong decoding baselines (Dola, AD) across hallucination detection, factual QA, and multi-choice reasoning. The overall pattern is that tasks requiring dispersed or nuanced evidence benefit the most, which aligns with our motivation: causal masking induces *semantic information solidification*, where early tokens disproportionately dominate attention, and our AMC-guided adjustment restores balance by reallocating focus to later context.

This trend is evident across different benchmarks. In the FACTOR datasets, particularly NEWS-FACTOR, long narrative contexts exacerbate prefix dominance: errors often arise from overweighting early story tokens while neglecting corrective evidence appearing later. By downweighting high-$S$ tokens, our method redistributes attention toward complementary information, explaining the larger gains observed in NEWS compared to WIKI-FACTOR, which relies more on local factual lookup and requires fewer long-range corrections. For HaluEval, which targets hallucination detection in short QA pairs, improvements are smaller in magnitude but remain consistent across model scales. Here, hallucinations usually stem from subtle question–evidence mismatches rather than long-range reasoning failures. Our adjustment reopens suppressed attention pathways in the second pass, allowing under-utilized tokens to re-enter consideration and improving detection even for smaller models such as LLaMA2-7B-chat. On TruthfulQA (MC), the improvements are most pronounced on MC3, which stresses nuanced factual precision under reasoning pressure. For example, on LLaMA3-8B-Instruct, MC3 rises from 28.9 (AD) to 33.9 (Ours). This pattern is consistent across model sizes: our adjustment maintains proportionality (samples with higher baseline scores remain higher after modification) while mitigating overconfidence in misleading tokens, thereby enhancing factual robustness.

From a scaling perspective, larger models such as LLaMA3-8B-Instruct show both higher baselines and larger gains, as their richer semantic associations carry a greater risk of amplifying prefix-dominated semantics. Our reweighting naturally counteracts this risk, leading to more pronounced improvements. Conversely, smaller models contain fewer dominant paths but still benefit from re-balancing, yielding smaller but steady improvements. Taken together, these observations show that the effectiveness of our method does not depend on dataset type, reasoning style, or model capacity,

but instead addresses a structural bias of autoregressive decoding. By combining AMC-based information flow estimation (Stage 1) with dynamic attention adjustment (Stage 2), our framework provides a task-agnostic solution that improves structured reasoning (FACTOR), hallucination detection (HaluEval), and nuanced factual QA (TruthfulQA MC). The cross-task consistency confirms that semantic solidification is a general phenomenon, and that mitigating it through principled reweighting leads to broad and reliable gains.

## 5.4 CASE STUDY VISUALIZATIONS

We conduct a case study on the HotpotQA dataset, chosen because its multi-hop structure is particularly prone to attention misallocation, the presence of distractor passages allows analysis of how irrelevant context is handled, and its overall complexity provides a rigorous testbed for evaluating attention modification. Figure 5 presents a representative token-level example where our method reshapes attention patterns: it selectively reduces information flow on dominant tokens while preserving semantic coherence across reasoning chains. This case provides intuitive evidence of how our framework mitigates semantic solidification, alleviating prefix-dominated biases and enabling more faithful reasoning over long contexts.

Figure 5: Case Study: Token-level analysis for a Kansas University question from HotpotQA. This example represents a *bridge-type* question requiring information synthesis across institutional and geographical contexts. **Top panel:** Original text with information scores before modification, where red intensity indicates higher scores for the top 40% of tokens. Key entities like "University of Kansas", "Lawrence", and "medical school" show high information scores. **Middle panel:** information score changes after RAS application, with blue intensity representing change magnitude for tokens with significant variations (top 40%). **Bottom panel:** Distribution comparison showing how SurFlow redistributes information scores across different value ranges.

# 6 CONCLUSION

We studied the problem of contextual faithfulness hallucinations in autoregressive LLMs and proposed a two-stage framework that models hidden-state evolution as an absorbing Markov chain combined with dynamic attention adjustment. This design enables retrospective integration of future semantics into earlier queries, mitigating prefix-induced bias without retraining or external resources. Experiments across FACTOR, HaluEval, and TruthfulQA benchmarks show consistent gains in factual accuracy and robustness, with especially strong improvements on long-range reasoning and nuanced factual precision. Overall, our work highlights semantic solidification as a structural source of hallucination and demonstrates that principled reweighting of attention provides a task-agnostic, efficient, and effective step toward building more reliable language models.

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

# A APPENDIX

## A.1 THEORETICAL DERIVATIONS

This section provides the theoretical foundation for RAS based on absorbing Markov chains.

**Absorbing Markov Chain: Detailed Derivation** An absorbing Markov chain is a Markov chain in which certain states, called absorbing states, cannot be left once entered. All other states are called transient states. In RAS, we use this framework to model the flow and correction of semantic information during decoding.

Suppose the Markov chain has $t$ transient states and $r$ absorbing states. The transition matrix $P$ can be written in canonical form:

$$P = \begin{pmatrix} Q & R \\ 0 & I_r \end{pmatrix} \tag{12}$$

where:

- $Q$ is a $t \times t$ matrix describing transitions among transient states.

- $R$ is a $t \times r$ matrix describing transitions from transient to absorbing states.

- $I_r$ is an $r \times r$ identity matrix for absorbing states.

**Fundamental Matrix and Limit Principle** The fundamental matrix $N$ is defined as:

$$N = (I_t - Q)^{-1} \tag{13}$$

where $I_t$ is the $t \times t$ identity matrix. The entry $N_{ij}$ gives the expected number of times the process is in transient state $j$ if it starts from transient state $i$.

The theoretical basis for $N$ comes from the following infinite sum:

$$N = I_t + Q + Q^2 + Q^3 + \cdots = \sum_{k=0}^{\infty} Q^k \tag{14}$$

This sum converges because all eigenvalues of $Q$ are less than 1 (since the chain is absorbing and will eventually leave transient states). Each term $Q^k$ represents the probability of being in each transient state after $k$ steps, starting from a given transient state. Thus, $N_{ij}$ is the expected total number of times the process visits state $j$ before absorption, starting from state $i$.

The convergence of the sum is guaranteed by the fact that as $k \to \infty$, $Q^k \to 0$ (the process is eventually absorbed). Therefore,

$$N = \lim_{n \to \infty} \sum_{k=0}^{n} Q^k = (I_t - Q)^{-1} \tag{15}$$

This is a standard result in matrix analysis for absorbing Markov chains.

**Expected Steps to Absorption** The expected number of steps before absorption, starting from transient state $i$, is:

$$t_i = \sum_{j=1}^{t} N_{ij} \tag{16}$$

**Absorption Probabilities** The matrix $B$ gives the probability of being absorbed in each absorbing state:

$$B = NR \tag{17}$$

where $B_{ij}$ is the probability that the process, starting from transient state $i$, is absorbed in absorbing state $j$.

| Method | TruthfulQA (MC) | | |
|---|---|---|---|
| | MC1 | MC2 | MC3 |
| Base Model | 33.60 | 51.30 | 24.90 |
| Ours w/o Col Sum | 33.54 | 51.84 | 25.05 |
| Ours w/o Info Score | 33.78 | 52.29 | 25.35 |
| Ours (Full) | **34.50** | **56.20** | **28.10** |

Table 2: Ablation study on TruthfulQA (MC) using LLaMA2-7B-chat, with different components removed.

## A.2 EXPERIMENTAL SETUP.

All experiments are conducted on NVIDIA 4090 GPUs with 48GB memory. For each dataset, we randomly sample 10% of the data as validation set for hyperparameter selection. For our approach, we search over $\alpha \in \{0.6, 0.7, 0.8, 0.9\}$ and $\beta \in \{0.8, 0.85, 0.9, 0.95\}$ for the experiment. Target layers are selected from $\{22, 24, 26, 28\}$ for 7B models and $\{30, 32, 34, 36\}$ for 13B models. The best configuration is selected based on validation performance.

## A.3 PSEUDOCODE

---
**Algorithm 1** Dynamic Two-Pass Corrective Attention
---
**Require:** Input tokens $x_{1:T}$, logits $\mathbf{z}$, parameters $\alpha, \beta$
**Ensure:** Corrected attention output $\tilde{O}$
  1: {Stage 1: Compute Information Score}
  2: Construct transition matrix $P$ from $\mathbf{z}$ and $x_{1:T}$
  3: Partition $P$ into transient $Q$ and absorbing $R$
  4: Compute fundamental matrix $N = (I - Q)^{-1}$
  5: Derive surprisal scores $s_j = -\log N_{1j}$
  6: {Stage 2: Adjust attention weights}
  7: For selected layers, remove causal mask to allow future context
  8: Adjust unmasked attention using $\mathbf{s}$ and utilization $\mathbf{c}$
  9: Fuse corrected attention output with original output using $\alpha, \beta$
10: **return** $\tilde{O}$

---

## A.4 ABLATION STUDY.

We conduct ablation experiments on the TruthfulQA (MC) dataset to analyze the contribution of each component.

**Effect of Each Component.** Table 2 summarizes the performance when removing or replacing major components. Removing the absorbing Markov chain (AMC) stage and directly applying dynamic attention with uniform token weights leads to a significant drop in MC2 and MC3, highlighting the necessity of accurate token-level semantic importance estimation. Similarly, removing the dynamic attention adjustment while keeping AMC also reduces performance, showing that identifying high-risk tokens alone is insufficient without actively modifying the attention pathways. These results confirm that both AMC and dynamic attention adjustment are indispensable to the proposed framework.

**Impact of Layer Selection.** We examine the effect of applying the second-pass attention adjustment to different Transformer layers. As shown in Figure 6, the gains gradually increase from shallow to middle layers, peaking at mid-to-upper layers (around layers 20–25), and then slightly drop at the final layers. This aligns with prior findings that factual knowledge and reasoning patterns are consolidated in higher layers, while adjustments at the final layers leave limited propagation steps for correction.

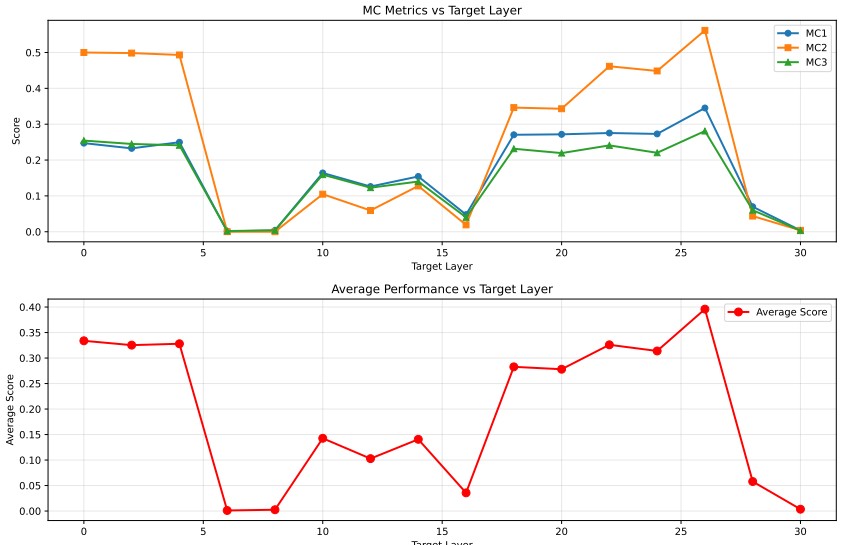

Figure 6: Impact of layer selection on TruthfulQA (MC) using LLaMA2-7B-chat.

**Impact of** $\alpha$**.** We vary $\alpha$ to study its effect on performance. Figure 7 shows that moderate $\alpha$ values (around 0.8) yield the best trade-off between factual accuracy and stability. Smaller values overly amplify the adjustment, leading to occasional semantic instability, while larger values preserve the original attention excessively, resulting in under-correction.

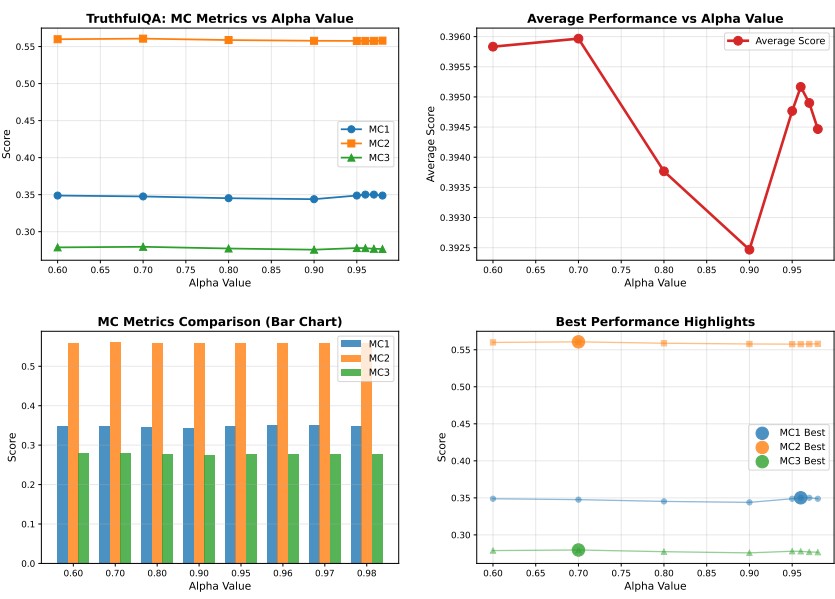

Figure 7: Impact of $\alpha$ on TruthfulQA (MC) using LLaMA2-7B-chat.

**Impact of** $\beta$**.** Figure 8 shows the effect of varying $\beta$, which controls the blending between masked and adjusted unmasked outputs. The optimal range is also around $\beta \approx 0.8$, similar to $\alpha$. Low $\beta$ values overly emphasize adjusted attention, which can destabilize output, while high $\beta$ values retain too much original attention, reducing the corrective effect.

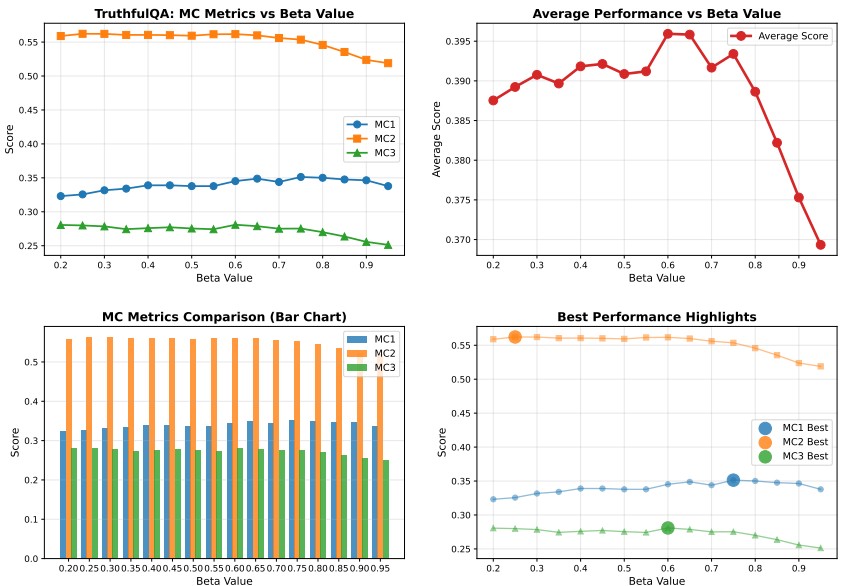

Figure 8: Impact of $\beta$ on TruthfulQA (MC) using LLaMA2-7B-chat.

## B ETHICS STATEMENT

This work follows the ICLR Code of Ethics. No human subjects or animal experiments were involved. All datasets used are publicly available and comply with their usage guidelines. No personally identifiable information was used, and the research does not raise privacy or security concerns.

## C REPRODUCIBILITY STATEMENT

We have taken care to ensure the reproducibility of our results. All code and datasets will be released upon acceptance. The paper provides detailed descriptions of the experimental setup, including training procedures, model configurations, and hardware specifications. We believe these measures will enable other researchers to reproduce our work and build upon it.

## D LLM USAGE

Large Language Models (LLMs) were used to improve the writing quality of this manuscript, including sentence rephrasing, grammar correction, and enhancing readability. The LLM was not involved in research design, methodology, data analysis, or the development of scientific ideas. Its contribution was limited to language refinement. The authors take full responsibility for the content and confirm that the use of the LLM complies with ethical standards.

