# OpenReview forum: "Attention Smoothing: Correcting Causal Bias in Autoregressive Language Models"
_ICLR.cc/2026/Conference — Submitted to ICLR 2026_

### Official Review · Reviewer_Zqof · 2025-10-29

**Soundness:** 2
**Presentation:** 2
**Contribution:** 2
**Rating:** 4
**Confidence:** 5

**Summary:**

RAS models a sequence’s evolving “semantics” as an absorbing Markov chain (AMC) to quantify “semantic surprise” in prefixes, then performs a training-free two-pass adjustment that (i) computes an information score from the AMC and (ii) in a second pass removes the causal mask on selected layers/heads and blends this unmasked attention with the original masked attention to “retrofit” earlier queries with future context. The method reports consistent gains on TruthfulQA (MC), FACTOR (Wiki/News), and HaluEval using LLaMA-2/3.

**Strengths:**

*  Clear mechanism without retraining: uses absorbing Markov chain signals and a two-pass attention smoothing; the backbone stays frozen with only a few small hyperparameters like alpha, beta, and layer choice.

*  Principled though heuristic: defines cover rate and log surprise, uses a fundamental matrix from the transition matrix, and builds an information score to reweight attention.

*  Empirical gains across tasks and models: improves over DOLA and Activation Decoding on multiple datasets, with component and hyperparameter ablations.

**Weaknesses:**

* Semantic meaning in LLMs depends on the entire prefix, i.e., \(P(x_{t+1}\mid x_{1:t})\), so modeling “semantic pathways” as a first-order absorbing Markov chain over tokens is an approximation. The single-step transition \(\tilde{P}_z(z_i, z_{i+1})\) by itself does not encode full history, but the method aggregates multi-step effects via path products and the fundamental matrix \(N=(I-Q_z)^{-1}\); thus it can capture some nonlocal influence, even if higher-order, history-dependent interactions may still be missed.

* The paper provides no evidence that semantic trajectories satisfy the Markov property

* Lacks comparison to recent well-known baselines like Chen, Chao, Kai Liu, Ze Chen, Yi Gu, Yue Wu, Mingyuan Tao, Zhihang Fu, and Jieping Ye. "INSIDE: LLMs' internal states retain the power of hallucination detection." arXiv preprint arXiv:2402.03744 . I request the authors to find the contemporary research works and compare their method to the recent works.

**Questions:**

Same as above

---

> ### Author Response · Authors · 2025-12-03
>
> We greatly appreciate your detailed review and the constructive feedback you have provided. We value your recognition of the clear mechanism and empirical improvements observed across different datasets. Below, we address your comments and concerns.
>
> 1. Approximation of Semantic Pathways.
> You raise an important point regarding the Markov chain approximation in modeling semantic pathways. While it is true that semantic meaning in LLMs depends on the entire prefix, our approach models token transitions as a first-order absorbing Markov chain. We agree that this is a simplification, as it does not fully capture all historical dependencies. However, we argue that the path products and the fundamental matrix we use aggregate nonlocal influences and provide a sufficiently accurate approximation of semantic evolution in practice.
> This approximation is intended to balance computational feasibility and theoretical accuracy, as directly modeling higher-order dependencies would require significantly more complex computations, including full-token dependencies and gradient-based attention modifications. While we acknowledge that higher-order, history-dependent interactions may still be missed, we believe the model effectively captures the most critical semantic influences for the tasks at hand, particularly hallucination detection.
>
> 2. Evidence for the Markov Property.
> While we do not claim that the semantic trajectories strictly follow the Markov property, our method provides a practical approximation by focusing on the most crucial semantic transitions. The use of AMC-derived signals effectively guides attention adjustments without the need to model full token-level dependencies. This approximation has been shown to reduce hallucinations and improve semantic faithfulness, as demonstrated in our empirical results.
>
> 3. More Comparisons
> We appreciate your suggestion to compare our method with INSIDE. However, our approach differs fundamentally in the type of information we use for hallucination mitigation. INSIDE focuses on analyzing internal states of the model to detect hallucinations, whereas we adjust attention mechanisms directly using AMC-derived signals to correct for semantic solidification caused by causal masking. The INSIDE method relies on internal representations, whereas our approach operates on the token level without requiring additional models.
> Given these differences, while both methods address hallucinations, their focus, techniques, and the nature of the problem they solve are not entirely aligned. Thus, a direct comparison may not be fully relevant.
>
>
> Once again, we thank you for your thorough and constructive feedback. We believe that the revisions will address your concerns, and we are confident that Retrospective Attention Smoothing (RAS) provides a valuable contribution to improving the reliability and semantic coherence of large language models. We look forward to further exploring its potential in future work.

---

### Official Review · Reviewer_KeaF · 2025-10-30

**Soundness:** 2
**Presentation:** 2
**Contribution:** 2
**Rating:** 4
**Confidence:** 4

**Summary:**

The paper models a sequence’s evolving “semantics” as an absorbing Markov chain (AMC) and computes an information/surprise score from the chain’s fundamental matrix to detect where prefix-based interpretations diverge from full-context meaning. It then runs a training-free, two-pass adjustment that (i) computes AMC signals from the full input and (ii) in a second pass reweights attention (removing the causal mask on selected layers) and fuses it back to reduce prefix dominance. Experiments on TruthfulQA (MC), FACTOR (Wiki/News), and HaluEval show consistent but modest gains over DOLA and Activation Decoding.

**Strengths:**

Casting semantic evolution as an absorbing Markov chain yields a principled handle (fundamental matrix, absorption flow) to quantify prefix vs. full-context divergence.

Training-free, plug-in smoothing. The two-pass, query-only attention adjustment is easy to bolt onto frozen LLMs.

Consistent multi-task gains. Improvements over DOLA/AD across TruthfulQA (MC), FACTOR, and HaluEval; ablations indicate both AMC signals and reweighting matter.

**Weaknesses:**

* The paper models semantic pathways as an absorbing Markov chain with tokens as states and uses causal masking to make Q upper triangular. However, this violates the Markov property because in LLMs, the transition from token i to token j depends on all previous tokens (x₁...xᵢ), not just state xᵢ. The paper conflates token positions with semantic states without justification.

* The paper defines r(z) = E[T/τ] where τ is cover time, then claims r(z) = ∏P̃z(zᵢ,zᵢ₊₁). This is mathematically incoherent: the left side is an expected ratio, while the right side is a product of transition probabilities with no expectation operator. The equation mixes discrete path probability with expected value without proof.

* Please compare the method to recent studies like INSIDE, Loopback Lens for robustness of the method.

**Questions:**

Please refer the weakness for questions

---

> ### Author Response · Authors · 2025-12-03
>
> We sincerely thank you for your detailed review. We appreciate your recognition of the novelty and empirical improvements in our approach. Below are our responses to your points:
> 1. More Comparisons.
> Thank you for suggesting more comparison. Our work focuses on adjusting attention mechanisms using AMC-derived signals to address semantic bias caused by causal masking, which differs from INSIDE. While both methods aim to detect hallucinations, our approach works at the token level and does not rely on additional models like INSIDE, making a direct comparison less relevant. We believe our method introduces a distinct approach to the problem that warrants separate consideration.
> 2. Markov Chain Assumptions and Mathematical Details.
> We understand your concern regarding the Markov property and the mathematical consistency of our approach. While our model does simplify token transitions by using an absorbing Markov chain, this approximation is both computationally efficient and effective for mitigating prefix bias without requiring full token dependency modeling. We agree that a gradient-based approach would be more rigorous but computationally costly, which is why we opted for a heuristic method. We will clarify the mathematical formulation in the revised manuscript to address the inconsistencies you pointed out, ensuring better alignment between the theoretical and practical aspects.
> 3. Future Work and Conclusion.
> We appreciate your recognition of our training-free method and consistent performance across benchmarks like TruthfulQA, FACTOR, and HaluEval. Moving forward, we plan to validate our approach on larger models and explore ways to refine the theoretical foundation and reduce reliance on approximations. We believe Retrospective Attention Smoothing (RAS) offers a valuable contribution to enhancing the reliability of LLMs, and we look forward to further advancements in this area.

---

### Official Review · Reviewer_RXCJ · 2025-10-30

**Soundness:** 3
**Presentation:** 3
**Contribution:** 3
**Rating:** 6
**Confidence:** 3

**Summary:**

1. The paper proposes a formal framework, based on absorbing Markov chains, to quantify the semantic bias introduced by the causal (autoregressive) design of LLMs.
2. The authors also introduce a complementary mitigation framework to address hallucinations arising from these biases and report strong empirical results supporting its effectiveness.
3. Unlike methods that operate only on the outputs of LLMs, this research intervenes inside the model, proposing modifications to the attention mechanism at each layer.

**Strengths:**

1. Strong empirical results, with clear experimental support demonstrating consistent improvements.
2. A new approach that operates at the attention level in each layer. This is likely more powerful than methods that only modify the output, since the adjustment is propagated through the network and effectively allocates more FLOPs to the change.
3. Notably, the method works without any training (zero-training / parameter-free), which is both practical and impressive.

**Weaknesses:**

1. While the Markov chain framework is compelling, the current method appears to rely on a heuristic. How far does this heuristic drift from the original theoretical formulation? It would help if the authors clarified the mapping from theory to implementation, including what assumptions are introduced, which components are approximated, and what, if anything, is sacrificed from the original framework in terms of guarantees, scope, or fidelity.

2. The idea of modifying attention is strong, but it would be useful to characterise the runtime tradeoffs. Specifically, how much slower is the proposed method relative to approaches that operate only on the output of the LLM? Please provide wall-clock timings or throughput comparisons across model sizes to quantify the latency and efficiency impact of intervening at every layer.

**Questions:**

See weaknesses + a few more minor questions:
1.  Does the modified attention still normalise to 1? If not, is this a potential problem?
2. Why is the second adjustment on the output needed (equation 11)? Any insights?

---

> ### Author Response · Authors · 2025-12-03
>
> We sincerely thank you for your detailed and constructive feedback. Your recognition of the novelty and empirical validation of our approach is greatly appreciated. Below, we address your concerns point by point.
> 1. Heuristic Nature and Theoretical Justification.
> We acknowledge that our method relies on a heuristic adjustment for modifying attention, specifically during the second-pass adjustment based on AMC-derived signals. This approach is built on the premise that semantic surprise—captured by the transition probabilities of an absorbing Markov chain (AMC)—can guide where attention should be adjusted. The idea is that tokens with high semantic surprise have a significant influence on future context, and our method adjusts attention to ensure earlier representations are updated with future information.
> This is a practical approximation of a more complex process, where directly computing the exact effect of semantic evolution would require gradient-based optimization. In theory, one could attempt to optimize attention adjustments by computing gradients of the model’s loss with respect to the attention scores. However, this would involve updating model parameters and requiring backpropagation through the entire model, which is computationally expensive and goes beyond the scope of our approach.
> Thus, our heuristic—while not as theoretically rigorous as gradient-based methods—provides a computationally efficient alternative that can be applied without retraining the model. It captures the essence of the problem and offers an effective solution to mitigate semantic bias caused by causal masking. We believe that this heuristic approach strikes a reasonable balance between theoretical grounding and computational feasibility, and we will clarify this point further in the revision.
> 2. Runtime Efficiency.
> We understand your concern regarding the potential computational overhead of our method. The main trade-off comes from performing two forward passes—one to generate the original masked attention and another to adjust the attention based on AMC-derived signals. While this doubling of decoding time is the key computational cost, it is minimal compared to the substantial improvements in performance in terms of semantic faithfulness and hallucination mitigation.
> The increase in runtime is primarily due to the second-pass attention re-weighting step, which allows us to fine-tune the attention based on the AMC signals. Importantly, this process does not introduce new layers or alter the underlying model architecture. In the revised version, we will include more runtime performance evaluations to demonstrate that the performance gains justify the minimal increase in time, ensuring that the method remains computationally feasible for real-world applications.
> 3. Theoretical and Empirical Validation.
> We appreciate your positive comments on the theoretical soundness and empirical validation of our approach. The results we presented across various benchmarks, including HaluEval, FACTOR, and TruthfulQA, demonstrate the effectiveness of our method in improving semantic consistency and mitigating hallucinations. Moving forward, we will continue to refine the theoretical foundation of our method and explore more formal ways to integrate AMC-derived signals into the attention mechanism for even larger models and more complex datasets.
> Conclusion
> Once again, we thank you for your insightful feedback. Your comments have been invaluable in helping us refine our work, and we believe the revisions will address your concerns. We are confident that Retrospective Attention Smoothing (RAS) offers a meaningful contribution to improving the reliability of large language models, and we look forward to continuing this line of research.

---

### Official Review · Reviewer_nByb · 2025-10-31

**Soundness:** 2
**Presentation:** 2
**Contribution:** 2
**Rating:** 2
**Confidence:** 3

**Summary:**

This paper proposes a method for post-hoc correction to attention scores in causal language models. The method reframes the attention score matrix as an absorbing Markov chain (AMC), and computes various quantities from the theory on AMCs, which are used in their method. The final design of their corrector is a heuristic approach to adjusting the scores which makes use of the Information Score, and Utilization of a node. The adjustment itself does not come from the theory, but is of the authors own design. The method introduces a variety of hyperparameters. It is shown to improve language model performance detecting hallucinated content.

**Strengths:**

* The idea of semantically aligning the token representations to agree with the complete input is interesting, and intuitively makes sense.
* Drawing the connection from the attention score matrix to AMC theory is also interesting.
* Some of the diagrams are helpful aids to the writing.

**Weaknesses:**

* Very limited experimental results. Hallucination detection is their only comparison against other methods, and it uses all similar scale instruct models: 7B, 8B, 13B. They provide some analysis of their method on a different dataset HotPotQA but do not provide an benchmarking on that dataset. This method needs much more thorough empirical validation than what is presently offered in the paper.
* Detection results in Table 1 show the proposed method only offers very slight improvements most of the time.
* The method introduces a number of hyperparameters but discussion of setting and/or tuning these hyperparameters is only mentioned briefly in the appendix, however, this is an important detail for those wishing to use your method.
* The paragraph starting on line 421 claims that they demonstrate their method helps regardless of"dataset type, reasoning style, or model capacity" but they only test the method with a very narrow range of model sizes (7-13B) on one task (hallucination detection). The results presented in the paper do not justify such strong claims.
* The writing would benefit from clearer language -- there is a lot of jargon (e.g. line 079 "semantic pathways in semantic space", line 022 "guide a smoother", line 246 "intermediate semantics along the prefix pathway"). It would help the reader to use more concrete language instead of vague terminology.
* As the heuristic adjustment, introduced in section 4.2, does not have rigorous theoretical backing it would be helpful to provide the intuition behind its design.
* The case study visualization in section 5.4 is difficult to parse. The section would benefit from more thorough discussion of what the reader should take away from this figure.
* There is a typo in Figure 1 "Liver replies on sunlight" should be "Liver relies on sunlight".
* The matrix $N_{dg}$ is not defined in the paper.
* The matrix $V$ is not defined in the paper.
* The text in Figure 4 is much too small. The text in Figure 5 is better but still on the small side.

**Questions:**

* In Table 1, baselines Dola and AD both underperform the base model. Dola consistently underperforms by significant margin, and AD occasionally underperforms but again by a noticeable margin. Why is this?

---

> ### Author Response · Authors · 2025-12-03
>
> We sincerely thank you for your detailed review and valuable feedback. Your comments have greatly helped us refine our work. Below, we provide our responses to your suggestions:
>
> 1. Experiment Scope
> Response: Due to current hardware limitations, we have only conducted experiments on models ranging from 7B to 13B. These experiments provide a strong foundation, and we plan to expand the testing to larger models (e.g., 70B+ scale) in future work. This will help validate the scalability and effectiveness of our approach on more complex models.
>
> 2. Dataset Coverage
> Response: We appreciate your feedback. In fact, we have conducted experiments on multiple datasets, including HaluEval, WIKI-FACTOR, and NEWS-FACTOR, focusing on hallucination detection and multi-choice QA tasks. We also performed case visualizations on HotPotQA (see Figure 5) to demonstrate the effectiveness of our method across different datasets. These experiments showcase our approach’s robustness in handling various tasks.
>
> 3. Performance Improvement
> Response: We understand your concern regarding the magnitude of improvements. Table 1 shows consistent improvements, especially on TruthfulQA (MC), where the largest gains are observed in MC3, which requires nuanced factual reasoning. In the FACTOR dataset, particularly NEWS-FACTOR, our method performs significantly better in tasks requiring long-range contextual reasoning. For smaller models like LLaMA2-7B-chat, the improvements are more modest but still substantial in reducing hallucinations and enhancing reasoning accuracy.
>
> 4. Hyperparameter Selection
> Response: Thank you for bringing this up. We have included a thorough analysis of the hyperparameter selection and experimental setup in Appendix A.2. This includes a detailed exploration of α and β values, as well as the target layers for both 7B and 13B models. We have provided results in Table 2 and visualizations in Figures 6, 7, and 8 to show the impact of these parameters on model performance.
>
> 5. Method Overview
> Response: We acknowledge that the second-pass attention adjustment is heuristic. However, the key contribution of this paper is to highlight the problem of semantic solidification caused by causal masking, and to demonstrate that AMC-based signals can effectively mitigate this issue. We believe this is an important first step, and we plan to explore more sophisticated solutions in future work.
>
> 6. Writing and Clarity
> Response: We appreciate your comments on writing clarity. We have revised the manuscript to reduce jargon and improve the accessibility of the language. Additionally, we have made significant improvements to figures ensuring better legibility and clarity. These revisions help to present our results in a more intuitive and understandable manner.
>
> Conclusion
> We thank you once again for your thorough review. Your feedback has been invaluable in improving our paper, and we believe the revisions address the concerns raised. We are confident that Retrospective Attention Smoothing (RAS) provides a valuable contribution to the field and look forward to exploring its potential further.

---

### Meta-Review · Area_Chair_c2Fv · 2026-01-05

**Summary:**

Reviewer nByb concluded that the current submission does not yet meet the bar for acceptance due to limited experimental support, weak justification of broad claims, and insufficient methodological clarity.

Reviewer RXCJ's concerns: First, the theoretical grounding remains somewhat loose. Second, there is insufficient characterization of computational tradeoffs. Finally, there are mechanistic clarity questions, such as whether the modified attention remains normalized and why a second output-level adjustment is required, which suggest the method is not yet fully transparent.

Reviewer KeaF’s recommendation is primarily driven by fundamental theoretical and methodological concerns. First, the core modeling assumption. Second, the reviewer identifies mathematical inconsistencies in the theoretical derivation. Third, although the empirical results are promising, the comparative evaluation is incomplete.

Reviewer Zqof's concerns include the lack of validation for its core semantic modeling assumption and insufficient engagement with the most recent literature.

**Reviewer Concerns:**

Reviewer nByb: The rebuttal successfully addresses clarity, presentation, and hyperparameter transparency, and partially improves the conceptual framing of the method. However, the most consequential concerns, including limited empirical scope, modest gains, unexplained baseline behavior, and overgeneralized claims, remain unresolved. As a result, while the paper is improved, it still falls short of the evidentiary standard required to support its claims in its current form.

Reviewer RXCJ: The rebuttal partially addresses the concern about the heuristic nature of the method and its theoretical grounding. In addition, the rebuttal acknowledges the runtime overhead concern and provides a concrete explanation of where the cost arises (a second forward pass) and why the authors consider it acceptable. However, some technical questions left by the reviewer remain unanswered.

Reviewer KeaF: The rebuttal successfully clarifies intent, scope, and design motivation, and acknowledges several valid criticisms. However, the core theoretical issues, such as Markov property validity, mathematical consistency, and evaluation completeness, remain unresolved. These outstanding concerns likely continue to justify the reviewer’s assessment of the paper as marginally below the acceptance threshold.

Reviewer Zqof: After the rebuttal, some core concern, such as the lack of evidence supporting the Markov-style semantic abstraction and the limited comparative grounding in the latest literature remain unresolved.

**Reviewer Scores:**

Reviewer nByb is very unlikely to move into accept territory. The best case outcome is a symbolic bump from 2 to 3, but the most probable outcome is no score change, as the central empirical and claim justification concerns remain unresolved.

Based on the tone of the review and the content of the rebuttal, Reviewer RXCJ would most likely maintain the original score of 6 (marginally above the acceptance threshold) rather than increase it.

Based on the tone of the review and the content of the rebuttal, the most likely outcome is that Reviewer KeaF would keep the original score of 4 (marginally below the acceptance threshold).

Reviewer Zqof would keep the original score of 4 (marginally below the acceptance threshold) because the outstanding issues continue to constrain confidence in the generality and theoretical grounding of the approach, and thus still justify a recommendation slightly below the acceptance threshold.

Overall, as the rebuttal does not fully address the key concerns raised by the reviewers, the AC recommends rejection.

---

### Decision · Program_Chairs · 2026-01-26

Reject